# CAR Triggered Release of Type-1 Interferon Limits CAR T-Cell Activities by an Artificial Negative Autocrine Loop

**DOI:** 10.3390/cells11233839

**Published:** 2022-11-30

**Authors:** Dennis Christoph Harrer, Charlotte Schenkel, Valerie Bezler, Marcell Kaljanac, Jordan Hartley, Markus Barden, Hong Pan, Astrid Holzinger, Wolfgang Herr, Hinrich Abken

**Affiliations:** 1Department Hematology and Internal Oncology, University Hospital Regensburg, 93053 Regensburg, Germany; 2Leibniz Institute for Immunotherapy, Division Genetic Immunotherapy, and Chair for Genetic Immunotherapy, University Regensburg, 93053 Regensburg, Germany

**Keywords:** CAR T-cells, TRUCK, inducible, negative feedback

## Abstract

The advent of chimeric antigen receptor (CAR) T cells expedited the field of cancer immunotherapy enabling durable remissions in patients with refractory hematological malignancies. T cells redirected for universal cytokine-mediated killing (TRUCKs), commonly referred to as “fourth generation” CAR T-cells, are designed to release engineered payloads upon CAR-induced T-cell activation. Building on the TRUCK technology, we aimed to generate CAR T-cells with a CAR-inducible artificial, self-limiting autocrine loop. To this end, we engineered CAR T-cells with CAR triggered secretion of type-1 interferons (IFNs). At baseline, IFNα and IFNβ CAR T-cells showed similar capacities in cytotoxicity and cytokine secretion compared to conventional CAR T-cells. However, under “stress” conditions of repetitive rounds of antigen stimulation using BxPC-3 pancreas carcinoma cells as targets, anti-tumor activity faded in later rounds while being fully active in destructing carcinoma cells during first rounds of stimulation. Mechanistically, the decline in activity was primarily based on type-1 IFN augmented CAR T-cell apoptosis, which was far less the case for CAR T-cells without IFN release. Such autocrine self-limiting loops can be used for applications where transient CAR T-cell activity and persistence upon target recognition is desired to avoid lasting toxicities.

## 1. Introduction

Over the last decade, chimeric antigen receptor (CAR) T-cells have become a relevant treatment option in the immunotherapy of hematological malignancies [1,2,3]. The efficacy of CAR T-cells against various entities is currently evaluated in a large body of clinical trials [4]. “Second-generation” CAR T-cells with both primary and costimulatory signaling domain exhibit robust anti-tumor activity owing to sustained proliferation and durable persistence [1]. For the treatment of a variety of solid cancer entities, CAR T-cell functionality is continuously under refinement [5]. While third generation CAR T-cells improve canonical T-cell functions by the integration of an additional co-stimulatory domain, fourth generation CAR T-cells, also referred to as “T cells redirected for universal cytokine-mediated killing” (TRUCKs), release engineered payloads upon CAR-triggered activation that act in an autocrine or paracrine fashion for modulating immune cell functions [6,7]. As one of the first examples, TRUCK T-cells were engineered to release IL-12, which is usually not produced by T cells, resulting in IL-12 mediated paracrine activation of macrophages and subsequent destruction of antigen-negative tumor cells [8]. The TRUCK concept basically allows for bestowing CAR T-cells with new functions mediated by produced payloads, which are not part of the canonic CAR T-cell secretome, such as IL-12, IL-18, and others [8,9]. While first TRUCKs were generated by transduction with two separate vectors encoding the CAR and the engineered payload, respectively [8], an “all-in-one” lentiviral vector system was recently established, relying on a single expression cassette for the constitutive expression of the CAR and the inducible expression of the payload driven by a modified NFATsyn promoter [10].

Current efforts are aiming at deliberately shutting down CAR T-cell activity, mostly in the case of toxicity, comprising drug-regulated or adaptor-based platforms. As examples, drugs are used to switch off CAR signaling [11], activate suicide genes [12], regulate CAR expression by controlling protease activities [13], and to stabilize ON/OFF-state confirmations of the CAR [14,15]. Addition or withdrawal of adaptor molecules enabling binding of CAR T-cells to respective target antigens represents an alternative approach of controlling CAR T-cell activity [16,17,18]. Finally, transient receptor transfer via mRNA electroporation constitutes a way to confine CAR T-cell persistence and CAR T-cell activity to 24–48 h [19,20].

For proof-of-concept, we exploited the TRUCK technology to generate CAR T-cells with an engineered negative-feedback loop based on CAR-activation-induced secretion of type-1 interferons (IFNs) that would allow full CAR-driven T-cell activation upon primary antigen contact and subsequent shut-down of activities in a CAR-triggered autocrine manner. The strategy is novel compared to existing approaches since it allows full CAR expression until target recognition; only repetitive target engagement will activate the IFN-mediated negative loop.

Physiologically, IFNs are immune stimulatory molecules that can equally foster innate and adaptive immune responses, restrict replication of viruses, and impair the growth of cancer cells [21,22,23,24,25]. With regard to T-cell functionality, however, IFNs assume a context-dependent dual role with promotion of T-cell cytotoxic functions on the one hand side [26,27,28,29], and restriction of T-cell proliferation and survival on the other side [30]. Moreover, high IFN-levels tend to eventuate in T-cell dysfunctionality and attrition [30] whereas low IFN levels favor T-cell functionality and persistence [26]. Given the importance of IFN dosing, we hypothesized that CAR-activation-induced production of medium IFN levels could sustain transient CAR T-cell functionality before IFN-mediated T-cell attrition would shut down CAR T-cell activity. Engineered, self-regulatory negative feedback loops to restrict long-term CAR T-cell activity are, to our best knowledge, novel and can be used for short-term CAR T-cell activities where CAR triggered T-cell amplification and long-term activity is not desired.

## 2. Materials and Methods

### 2.1. Cells and Reagents

Peripheral blood mononuclear cells (PBMCs) were purified from blood from healthy donors upon informed consent and approval by the institutional review board (21-2224-101 Regensburg) using density centrifugation on Lymphoprep (Axis-Shield, Oslo, Norway). T cells were cultured in RPMI 1640 medium supplemented with GlutaMAX (Gibco, ThermoFisher, Waltham, MA, USA), 100 IU/mL penicillin, 100 µg/mL streptomycin (Pan-Biotech, Aidenbach, Germany), 2 mM HEPES (PAA, GE healthcare, Chicago, IL, USA) and 10% (*v*/*v*) heat-inactivated fetal calf serum (Pan-Biotech, Aidenbach, Germany). The 293T is a human embryonic kidney cell line that expresses the SV40 large T antigen. BxPC-3 is a human pancreatic cancer cell line (ATCC CRL-1687; American Type Culture Collection, Manassas, VA, USA). The cells were cultured in DMEM supplemented with GlutaMAX (Gibco, ThermoFisher, Waltham, MA, USA), 100 IU/mL penicillin, 100 µg/mL streptomycin (Pan-Biotech, Aidenbach, Germany), 2 mM HEPES (PAA, GE healthcare, Chicago, IL, USA) and 10% (*v*/*v*) heat-inactivated fetal calf serum (Pan-Biotech, Aidenbach, Germany).

### 2.2. CAR T-Cell Generation

Cryopreserved PBMCs were thawed and activated same day with anti-CD3 mAb OKT-3, CD28 mAb 15E8 and IL-2 (1000 IU/mL). IL-2 (200 IU/mL) was added on days 2, 3, and 4 after activation. Retroviral transduction was performed on days 2 and 3 after activation. Tissue culture T25 flasks were coated overnight at 4 °C with 2 mL Poly-L-Lysine (10 μg/mL) (Sigma-Aldrich, St. Louis, MO, USA). Thereafter, T25 flasks were rinsed, and fresh retroviral supernatant was added at 5 mL per flask and centrifuged for 30 min at 32 °C at 3000× *g*. Then, T cells were resuspended in 5 mL retroviral supernatant, added to the flasks and centrifuged for 90 min at 32 °C at 1600× *g*. The retroviral supernatant was produced in 293T cells after “PEIpro” (Polyplus, Illkirch, France) mediated transfection as previously described [31]. The expression cassette encoding the CEA-specific CAR BW431/26scFv-Fc-CD28-ζ was described earlier [32]. The vectors encoding CEA-specific CAR T-cell with CAR-induced IFNα2 secretion (BW431/26scFv-Fc-CD28-ζ-NFAT-IFNα2) or CAR-induced IFNβ secretion (BW431/26scFv-Fc-CD28-ζ-NFAT-IFNβ) were obtained by insertion of the NFATsyn promoter [10] followed by IFNα or IFNβ encoding genes (synthesized by GenScript Biotech, New Jersey, NJ, USA) into the BW431/26scFv-Fc-CD28-ζ expression vector. Four days after activation, CAR T-cells were enriched by magnetic activated cell sorting (MACS) using a biotinylated goat F(ab’)2 anti-human IgG antibody (Southern Biotech, Birmingham, AL, USA) followed by purification with anti-biotin micro-beads (Miltenyi Biotec, Bergisch Gladbach, Germany). CAR T-cells were used for in vitro assays after a 24 h culture period in IL-2 free medium.

### 2.3. Flow Cytometry

Surface staining was carried out by incubating cells with antibodies at 4 °C for 15 min. For intracellular staining, cells were fixed and permeabilized with “Transcription Buffer” set (BD Biosciences, Franklin Lakes, NJ, USA) for 30 min at 4 °C. The fixable viability dye eFluor 780 (ThermoFisher) was employed to exclude dead cells from analysis. Fluorescent-minus-one (FMO) controls were used to set gates. The following antibodies used for flow cytometry were purchased from Southern Biotech: goat F(ab’)2 anti-human IgG-PE and goat F(ab’)2 anti-human IgG-FITC to detect CAR expression. The antibodies were FITC-conjugated anti-CD3 (clone BW 264/56), APC-conjugated anti-Granzyme B (clone REA 226), and APC Vio770-conjugated anti-CD66abcde (clone TET2) were obtained from Miltenyi Biotec. The following antibodies were purchased from Biolegends (San Diego, CA, USA): BV605-conjugated anti-PD1 (clone EH12.2H7) and BV421-conjugated anti-Ki67 (clone Ki-67). The following antibodies were purchased from BD Biosciences (Franklin Lakes, NJ, USA): BV421-conjugated anti-TIM3 (clone 7D3) and PerCP-Cy5.5-conjugated anti-LAG3 (11C3C65). Annexin V staining was performed with PE “Annexin V Apoptosis Detection Kit” (BD Biosciences) according to the instructions of the manufacturer. Immunofluorescence was measured using a BD FACSLyric (BD Biosciences) equipped with FACSuite software (BD Biosciences). Data were analyzed using FlowJo software version 10.7.1 Express 5 (BD Biosciences).

### 2.4. Cytokine Secretion

CEA-negative 293T cells and CEA-positive BxPC-3 cells were seeded in 96-well round-bottom plates (1 × 10^5^ cells per well) overnight; un-transduced T cells or CAR modified T cells were added at 1 × 10^5^ cells per well in 200 μL medium. Supernatants were harvested after 48 h of co-culture. Subsequently, IL-2 and IFNγ in culture supernatants were detected by ELISA as previously described [33]. Analysis of type-1 IFNs in the supernatant was performed using the rapid bioluminescent human IFN-α2 and IFN-β ELISA kit (InvivoGen, San Diego, CA, USA).

### 2.5. Bioactivity of IFNs

The bioactivity of CAR-activation-induced IFNα2 or IFNβ was measured using human IFN reporter HEK 293 cells (InvivoGen, San Diego, CA, USA) according to the manufacturer’s instructions. Supernatants were harvested after 48 h and added to reporter HEK cells. Of note, freshly thawed reporter HEK cells were cultured for two weeks before experimental use. Recombinant IFNα (Sigma-Aldrich, St. Louis, MO, USA) was added as positive control.

### 2.6. Cytotoxicity Assay

CAR T-cells (0.125–10 × 10^4^ cells/well) were co-cultivated for 24 h in 96-well round-bottom plates with CEA-positive BxPC-3 cells or CEA-negative 293T cells (each 1 × 10^4^ cells/well). Specific cytotoxicity of CAR T-cells was monitored by a XTT-based colorimetric assay employing using the “Cell Proliferation Kit II” (Roche Diagnostics, Mannheim, Germany). Viability of tumor cells was calculated as mean values of six wells containing only tumor cells subtracted by the mean background level of wells containing medium only. Non-specific formation of formazan due to the presence of T cells was determined from triplicate wells containing T cells in the same number as in the corresponding experimental wells. The number of viable tumor cells in experimental wells was calculated as follows: viability (%) = [OD(experimental wells − corresponding number of T cells)]/[OD(tumor cells without T cells − medium)] × 100. Cytotoxicity (%) was defined as 100 − viability (%).

### 2.7. Repetitive Stimulation Assay

GFP engineered BxPC-3 cells were seeded in 12 well plates at a density of 0.1 × 10^6^ cells per well. After 24 h, CAR T-cells were added (0.1 × 10^6^ cells/per well). After the first round of stimulation (3 days), all cells were harvested, washed with PBS and re-suspended in 1 mL T-cell medium. Finally, 100 μL were used for counting of live GFP+ tumor cells and CAR T-cells by flow cytometry using counting beads (CountBright, ThermoFisher). The remaining 900 μL were added to a fresh 12-well plate with 0.1 × 10^6^ BxPC-3 cells for four days i.e., Round 2 of stimulation. This procedure was reiterated in Round 3 and Round 4.

### 2.8. Statistical Analysis

Statistical analysis was performed with GraphPad Prism, Version 9 (GraphPad Soft-ware, San Diego, CA, USA). P values were calculated by Student’s *t* test; ns indicates not significant, * indicates *p* ≤ 0.05, and ** indicates *p* ≤ 0.01.

## 3. Results

### 3.1. Development of CAR T-Cells with CAR Triggered Release of Bioactive IFN-α or IFN-β

In order to engineer CAR T-cells with CAR-activation-induced secretion of type-1 IFN (IFN-CAR T-cells), we designed a novel retroviral expression cassette encoding a carcinoembryonic antigen (CEA) specific CAR (CEA-28ζ) for followed by a synthetic NFATsyn promoter controlling the expression of either IFNα (CEA-28ζ-IFNα) or IFNβ (CEA-28ζ-IFNβ). The NFAT6 enhancer is activated upon CAR signaling to trigger the transcription of the engineered payload. Human T cells were retrovirally transduced with the respective expression cassettes depicted in Figure 1A. Whereas the anti-CEA CARs with and without transgenic IFNα were equally expressed, CAR T-cells with IFNβ exhibited slightly higher CAR expression levels (Figure 1B). For the following analyses, CAR-positive cells were enriched using magnetic-activated cell sorting (MACS). To verify CAR-triggered release of type-1 IFN, CAR T-cells were co-incubated with CEA-positive BxPC-3 pancreatic cells; CEA-negative 293T cells served as control (Figure 1C). IFNα-CAR T-cells released IFNα upon CAR activation and IFNβ-CAR T-cells released IFNβ while CAR T-cells without payload did not show IFN production as expected (Figure 1D). Secreted IFNα and IFNβ were biologically active as indicated by the HEK Blue human IFN reporter cell line that responds to IFN receptor signaling (Figure 1E). Supernatants of IFNβ CAR T-cells evinced the greatest molar bioactivity, which is in line with the fact that IFNβ has higher affinity for the IFN receptor as compared to IFNα.

Taken together, data indicate that CAR engagement of cognate antigen triggers the release of transgenic, bio-active IFNα and IFNβ, respectively.

### 3.2. IFN-CAR T-Cells and Conventional CAR T-Cells Showed Similar Cytotoxicity and Cytokine Secretion

We compared cytotoxicity of IFN-CAR T-cells with CAR T-cells without payload in a 24 h in vitro killing assay. To this end, CAR T-cells were co-incubated with CEA-negative 293T cells and CEA-positive BxPC-3 pancreatic cells. Across different effector-to-target ratios, CAR T-cells with and without IFN payload displayed equally robust killing of BxPC-3 cells without significant differences between the CAR constructs; CEA-negative 293T cells were not lysed (Figure 2A). Upon antigen-specific stimulation with BxPC-3 cells, IFN-CAR T-cells and CAR T-cells accumulated similar concentrations of IL-2 and IFNγ in the culture supernatants indicating CAR T-cell activation; only background cytokine release was observed without antigen-specific stimulation (Figure 2B). T cells without a CAR did not show increase in cytokine production.

Taken together, IFN-CAR T-cells and conventional CAR T-cells displayed similar cytotoxicity and cytokine secretion in a short-term cytotoxicity assay.

### 3.3. IFN-CAR T-Cells Exhibited Self-Limiting Activity upon CAR Signaling

We aimed at using CAR-induced IFNα or IFNβ release to limit CAR-redirected T-cell activation in an autocrine manner after some rounds of antigen engagement. To address the issue, we subjected CAR T-cells to an in vitro repetitive stimulation assay (Figure 3A). IFN-CAR T-cells and CAR T-cells without IFN payload were co-incubated with GFP-labeled CEA+ BxPC-3 cells for four consecutive rounds (Round 1–4), each round of stimulation lasting three to four days. At the end of each round, CAR T-cells and tumor cells were counted by flow cytometry using counting beads. Starting by seeding 1 × 10^5^ CAR T-cells, CAR T-cells initially expanded within the first two rounds of stimulation; however, steadily declined over the following rounds (Figure 3B). Similarly, CAR T-cells with IFNα showed an expansion phase during the first two rounds, and a contraction phase in the last two rounds. Although not reaching significance, IFNα CAR T-cells seemed to show weaker expansion and limited persistence as compared to CAR T-cells (Figure 3B). On the contrary, CAR T-cells with IFNβ release did not display substantial T-cell expansion in the first round, and T-cell counts rapidly disappeared in the later rounds (Figure 3B). In terms of anti-tumor activity, both CAR T-cells and IFN-CAR T-cells exhibited robust elimination of BxPC-3 cells in the first two rounds of stimulation while the cytolytic activity was extinguished in round three (Figure 3C). For comparison, conventional CAR T-cells were still capable of eliminating 50% of seeded BxPC-3 cells in the final round. Data demonstrate that CAR T-cells with induced release of IFNα and particularly IFNβ decline in number and anti-cancer cell activity under repetitive stimulatory conditions, which is far less the case for canonical CAR T-cells without IFN release. We concluded that IFN-CAR T-cells exhibit self-limiting activity once stimulated through their CAR.

To address the underlying mechanism, we monitored apoptosis, proliferation, granzyme B expression, and markers associated with T-cell exhaustion during repetitive stimulation with CEA+ BxPC-3 cells. At the end of all three rounds of cancer cell engagement, IFNβ-CAR T-cells showed a significantly greater upregulation of Annexin V indicating enhanced rate of apoptosis compared to CAR T-cells and IFNα-CAR T-cells (Figure 4B). Without CAR stimulation, however, the frequency of apoptotic cells was the same for IFNα-CAR, IFNβ-CAR and canonical CAR T-cells. In contrast, the proliferative capacity of the CAR T-cells initially did not significantly differ; however, capacity substantially dropped after the first round in all conditions as measured by Ki67 expression (Figure 4C). We concluded that the lack of expansion and early contraction of IFNβ-CAR T-cells is likely due to enhanced apoptosis rather than to inferior proliferation. The load of the cytotoxic effector molecule granzyme B was similar in CAR T-cells and IFN-CAR T-cells (Figure 4D) ruling out that lack of cytolytic granules caused decline in anti-cancer cell activities. After the first stimulation round, IFNβ-CAR T-cells, moreover, showed a higher expression of molecules associated with exhaustion, such as PD-1, LAG-3, and TIM-3 (Figure 4E–G).

Collectively, IFNβ-CAR T-cells showed enhanced apoptosis after CAR stimulation, building the mechanistic basis for their self-limiting persistence and function.

## 4. Discussion

We aimed at engineering CAR T-cells with self-limiting capacities upon CAR engagement of the target. The aim was realized by adding an IFN-based negative feedback loop to CAR T-cells that shuts down CAR T-cell activity in an autocrine manner once the CAR repetitively triggers T-cell activation. For engineering we used the novel modular “all-in-one” vector system that links constitutive CAR expression with the inducible expression of engineered payloads in a retroviral system. IFN-CAR T-cells release the respective transgenic type-1 IFN upon CAR stimulation; also, these cells displayed cytotoxicity and cytokine secretion at the same degree as conventional CAR T-cells upon first antigen contact. Upon repetitive stimulation through their CAR, however, IFNβ-CAR T-cells exhibited pronounced self-limiting activity and persistence due to enhanced apoptosis and exhaustion. To our best knowledge, this is the first study to establish an autocrine negative feedback loop for CAR T-cells which gets into action particularly after CAR stimulation; without CAR engagement of target, IFN-CAR T-cells persist in the same manner as CAR T-cells without IFN payload.

IFNα and IFNβ released by engineered CAR T-cells, in addition, can potentially sustain the anti-tumor attack of CAR T-cells, in particular by blocking proliferation and induction of apoptosis in malignant cells, destruction of tumor-associated blood vessels, increase of MHC expression on malignant cells, promotion of dendritic cell and macrophage functionality, and activation of NK cells [34]. While IFNα and IFNβ exhibit similar activities, IFNβ shows higher receptor affinity, resulting in stronger IFN-mediated effects that result in stronger induction of apoptosis [35]. Along this line, self-limiting CAR triggered T-cell activation was more pronounced in IFNβ-CAR T-cells than in IFNα-CAR T-cells due to the enhanced induction of apoptosis. 

The IFNβ effect is dose-dependent; CAR T-cells secreting low IFNβ levels (<10 pg/mL) show superior therapeutic efficacy due to balanced tumoricidal function, increased T-cell persistence, and decreased exhaustion [26]. Low level IFNβ secretion was achieved by combined signaling of a CD28-ζ CAR and enforced expression of 4-1BBL. In contrast, high IFNβ levels (>10 ng/mL) resulted in a pronounced attrition of CAR T-cells incapable to extend the survival of tumor-bearing mice [30]. Here, we are taking advantage of the dose-dependent IFNβ effect; a first round of stimulation and thereby low IFNβ levels barely impact CAR T-cell survival and anti-tumor activity, while repetitive rounds increase IFNβ release to levels that induce CAR T-cell apoptosis. Intermediate IFNβ levels (0.1 to 1 ng/mL) are likely ideal to balance the IFN-based feedback loop on CAR T-cell activity. Finally, higher levels of IFNβ could stir more IFN-related side effects, such as fever, malaise, and myalgia [36].

Alternative strategies to achieve short-term CAR T-cell activity encompass drug-regulated platforms, adaptor-based platforms, and mRNA electroporation. In contrast to the infusion of drugs or adaptor molecules, mRNA electroporation creates CAR T-cells with intrinsic temporally limited activity due to transient CAR expression. However, CAR expression in RNA CAR T-cells is usually very short-lived restricting CAR T-cell activity to a few days after genetic engineering [20]. In the present study, we engineered an IFN-based negative feedback loop that is only initiated upon CAR triggering and independent from the addition of drugs; without CAR triggering, no IFNα or IFNβ is released and IFN-CAR T-cells are capable to persist.

The IFN-CAR T-cells with self-limiting activities can be used in short-term CAR T-cell applications where CAR T-cell persistence or long-term CAR T-cell activity is not desired. Short-term CAR T-cell activity could be exploited in neo-adjuvant therapies paving the way for surgical removal of tumors. In pancreatic cancer, colon cancer or other solid tumors, type-1 IFNs could not only act as a negative-feedback loop but also exert direct anti-tumor activities [37,38]. Moreover, IFN-CAR T-cells targeting CD19 could be used during the lymphodepletion step preceding the infusion of a CAR T-cell product of any specificity. In parallel to the physiological role of IFN in viral infections to execute endogenous lymphodepletion [39], CAR-induced IFNs could booster lymphodepletion, inhibit tumor growth, and promote stimulation of immune cells. In particular, engineered CAR T-cells with self-limiting persistence could target molecules (e.g., antigens with co-expression on healthy tissue) that would cause lasting toxicities once targeted by conventional CAR T-cells. Furthermore, promising target antigens, such as CD30 in Hodgkin´s lymphoma and colon cancer [33,39], or CD5 on T-cell lymphoma [40], are co-expressed by healthy T cells creating the risk of durable damage to endogenous healthy T-cells upon targeting by conventional CAR T-cells. Finally, CAR T-cells have gained attraction beyond the realm of oncology and are currently being evaluated as a promising therapy for autoimmune diseases, such as systemic lupus erythematosus [41]. Here, CAR T-cells with self-limiting activities could lead to depletion of auto-reactive B-cells in a short-term CAR T-cell therapy without the need for potentially life-long persistence of genetic cell products in non-tumor patients.

An essential step prior to clinical translation is the exploration of IFN-CAR T-cells in murine tumor models. In order to analyze the potential of IFN-CAR T-cells to act as a short-term neo-adjuvant therapy intended to reduce tumor size for subsequent surgical removal, we propose an orthotopic model of pancreas carcinoma. In this model, transplanted pancreatic carcinoma cells progressively fill up the abdominal cavity within three weeks [9]. Upon injection of IFN-CAR T-cells at various time points after tumor implantation, the power of this novel CAR T-cell construct to serve as a neo-adjuvant CAR T-cell therapy with transient activity can be investigated. Transient CAR T-cell activity can be monitored by declining CAR T-cell presence in blood and tumor tissue. Another suitable model for confirming the short-term activity of IFN-CAR T-cells is represented by the NALM6 leukemia model. Contrary to conventional CAR T-cells, which rapidly expand in the NALM6 model, IFN-CAR T-cells should exert anti-leukemia activity before showing swift contraction. Importantly, the NALM6 model is well-suited to study the re-emergence of tumor cells after initial successful tumor clearance [26]. As prolonged persistence of CAR T-cells is vital for durable complete remissions, the temporally restricted activity of IFN-CAR T-cells could be used for short-term operations preceding definitive tumor clearance. In case of swift tumor cell re-emergence after IFN-CAR T-cell therapy, another injection of IFN-CAR T-cells could be conceived. However, a second re-emergence of tumor cells should be treated with long-lasting conventional CAR T-cells to augment the chances for complete eradication of tumor cells.

## Figures and Tables

**Figure 1 cells-11-03839-f001:**
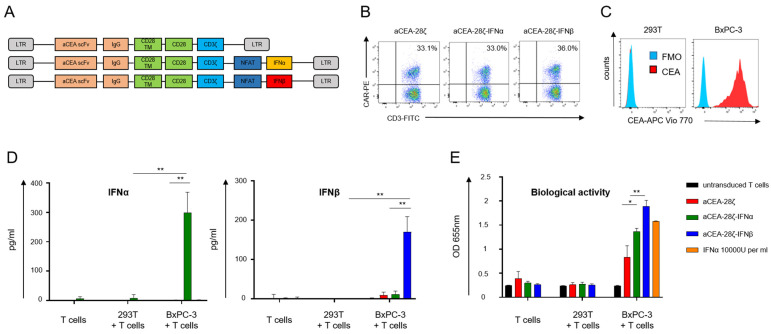
CAR-signaling induced secretion of bio-active IFN. (**A**) Schematic depiction of anti-CEA CAR constructs without and with CAR-inducible IFN expression cassette. (**B**) Flow cytometric analysis of CAR expression in T cells as detected by a Phycoerythrin (PE)-labeled anti-IgG antibody that recognizes the common spacer in the CAR exodomain. One representative donor out of five is shown. (**C**) Staining of target cell lines 293T and BxPC-3 for CEA expression using an APC Vio 770-conjugated anti-CEA antibody. One representative out of three independent experiments is shown. (**D**) ELISA-based quantification of IFNα and IFNβ in the supernatants after 48 h co-culture of engineered CAR T-cells without target cells and with CEA− 293T cells and CEA+ BxPC-3 cells. Data represent means ± SEM of three T-cell donors, *p* values were calculated by Student’s *t* test, ** indicates *p* ≤ 0.01. (**E**) HEK Blue reporter cell line assay to indicate the bioactivity of released IFN in the culture supernatant after 48 h co-incubation of CAR T-cells without and with 293T cells or BxPC-3 cells. Recombinant IFNα was used as positive control. Data represent means ± SEM of three donors, *p* values were calculated by Student’s *t* test, * indicates *p* ≤ 0.05, ** indicates *p* ≤ 0.01.

**Figure 2 cells-11-03839-f002:**
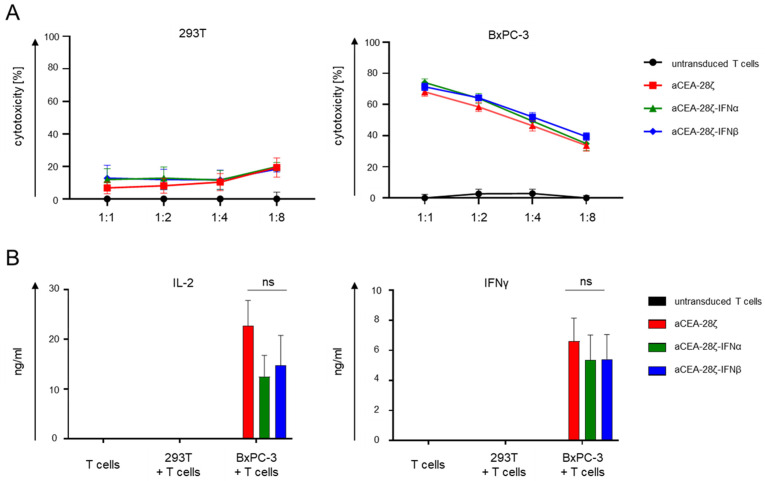
Upon first activation, IFN-CAR and conventional CAR T-cells show similar cytotoxicity and cytokine secretion. (**A**) Cytotoxicity of CAR T-cells was recorded upon a 24 h co-culture with 293T cells or BxPC-3 cells at the indicated effector-to-target ratios. Data represent means ± SEM of three donors, *p* values were calculated by Student’s *t* test, ns indicates not significant. (**B**) ELISA-based quantification of IL-2 and IFNγ accumulated in the culture supernatant after 48-h co-culture of CAR T-cells without or with 293T cells or BxPC-3 cells. Data represent means ± SEM of three donors, *p* values were calculated by Student’s *t* test, ns indicates not significant.

**Figure 3 cells-11-03839-f003:**
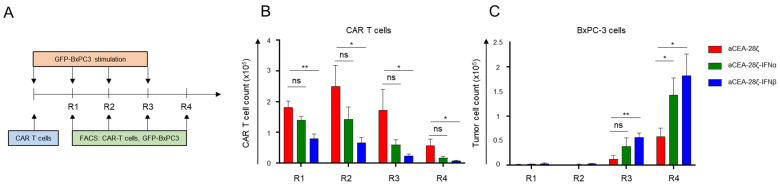
Self-limiting activity and persistence of IFN-CAR T-cells. (**A**) Experimental design of the repetitive antigen-stimulation assay. CAR T-cells (starting with 1 × 10^5^ CAR T-cells) with and without IFN payload were subjected to four rounds (R1–R4) of antigen-stimulation with GFP-labeled CEA+ BxPC-3 cells (1 × 10^5^ tumor cells at start of each round). At the end of each round, CAR T-cells (live CD3+/CAR+) (**B**) and BxPC-3 cells (live GFP+ tumor cells) (**C**) were counted by flow cytometry using counting beads. (**B**,**C**) Data represent means ± SEM of nine T-cell donors, *p* values were calculated by Student’s *t* test, ns indicates not significant * indicates *p* ≤ 0.05, ** indicates *p* ≤ 0.01.

**Figure 4 cells-11-03839-f004:**
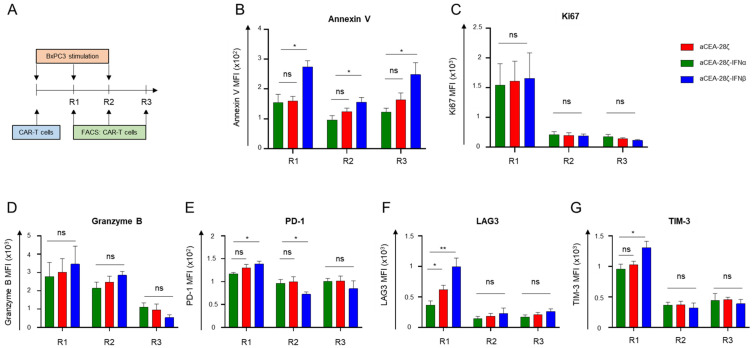
Enhanced apoptosis in IFNβ-CAR T-cells. (**A**) Experimental design of the repetitive antigen-stimulation assay. CAR T-cells were subjected to three rounds (R1–R3) of stimulation with CEA+ BxPC-3 cells. At the end of each round, CAR T-cells were stained for Annexin V (**B**), Ki67 (**C**), Granzyme B (**D**), PD-1 (**E**), LAG3 (**F**), and TIM-3 (**G**). Data represent geometric means of ± SEM of four donors, *p* values were calculated by Student’s *t* test, ns indicates not significant * indicates *p* ≤ 0.05, ** indicates *p* ≤ 0.01.

## Data Availability

Data are available upon reasonable request from the first author.

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
