# Peer review of "CAR Triggered Release of Type-1 Interferon Limits CAR T-Cell Activities by an Artificial Negative Autocrine Loop"

_cells, 2022, doi:10.3390/cells11233839_

Round 1

Reviewer 1 Report

The manuscript should be closely edited and revised to correct the many references not cited in the text including  5, ,34, 36  and 40-42. There are several typographical errors 105 should be 105 etc. 

  1. What is the main question addressed by the research? 

Improving utility of CAR T cells for therapeutic efficacy.

2. Do you consider the topic original or relevant in the field? Does it 
address a specific gap in the field? 

This is a novel area of development 

3. What does it add to the subject area compared with other published 
material? 

Increasing the payload of CAR T cells can also provide for a self limitation 

4. What specific improvements should the authors consider regarding the 
methodology? What further controls should be considered? 

In vivo assessment would be most appropriate

5. Are the conclusions consistent with the evidence and arguments presented 
and do they address the main question posed? 

Yes but the context of any potential to contribute to increased efficacy of tumour treatment requires in vivo assessment at least.

6. Are the references appropriate? 

Mostly but many are not cited and this needs to be corrected 

Author Response

We thank the reviewer for taking the time to review our manucript.

We apologize for the uncited references. We linked all uncited references to the corresponding text. 

Ref 5. --> line 34

Ref. 34 --> line 323

Ref. 36 --> line 339, the corresponding text about IFN-mediated side-effects was inserted

Ref. 40, 41 --> line 362

Ref 43 --> line 43

Moreover, we apologize for typographical errors. We scrutinized the manuscript and corrected remaining errors. 

Reviewer 2 Report

The authors have presented the model of self limiting CAR T cell in teh field of cellular immunotherapy towards solid tumors, importantly with the expression of INF, these cells become more potent as well as the CAR T cells will not persist . The invitro study provides a good model for researchers to implement into animals and then for clinical approaches. 

The authors should discuss how this study may show potential in animal models of Tumor as well as proposing a tumor model for clinical applications. The authors should discuss the proposition of how the reemergence of tumors even after therapy is likely why genetically engineered CAR T cell persistance is required. Does the patient in that case keep requiring a reinfusion of CAR T cells?

Author Response

Thank you for taking the time to review our manuscript. 

Indeed, the evaluation of IFN-CAR T cells in animal model is vital for clinical translation. We proposed an orhtotopic pancreatic model to serve a a good testing environment to assess the power of IFN CAR T cells to serve as a neo-adjuvant therapy with transient activity. Moreover, we discussed the use of the NALM6 leukemia model to confirm transient CAR T cell activity. 

In case of tumor reemergence, we advocate re-infusion of IFN-CAR T cells. But a second re-emergence should be treated with conventional CAR T cells. 

We added a an additional paragraph to the discussion section.